# Predicting the Association of Metabolites with Both Pathway Categories and Individual Pathways

**DOI:** 10.3390/metabo14090510

**Published:** 2024-09-21

**Authors:** Erik D. Huckvale, Hunter N. B. Moseley

**Affiliations:** 1Markey Cancer Center, University of Kentucky, Lexington, KY 40536, USA; edhu227@g.uky.edu; 2Superfund Research Center, University of Kentucky, Lexington, KY 40536, USA; 3Department of Toxicology and Cancer Biology, University of Kentucky, Lexington, KY 40536, USA; 4Department of Molecular and Cellular Biochemistry, University of Kentucky, Lexington, KY 40536, USA; 5Institute for Biomedical Informatics, University of Kentucky, Lexington, KY 40536, USA

**Keywords:** metabolism, pathways, biochemistry, machine learning, binary classification, multi-layer perceptron, metabolites, supervised learning, transfer learning

## Abstract

Metabolism is a network of chemical reactions that sustain cellular life. Parts of this metabolic network are defined as metabolic pathways containing specific biochemical reactions. Products and reactants of these reactions are called metabolites, which are associated with certain human-defined metabolic pathways. Metabolic knowledgebases, such as the Kyoto Encyclopedia of Gene and Genomes (KEGG) contain metabolites, reactions, and pathway annotations; however, such resources are incomplete due to current limits of metabolic knowledge. To fill in missing metabolite pathway annotations, past machine learning models showed some success at predicting the KEGG Level 2 pathway category involvement of metabolites based on their chemical structure. Here, we present the first machine learning model to predict metabolite association to more granular KEGG Level 3 metabolic pathways. We used a feature and dataset engineering approach to generate over one million metabolite-pathway entries in the dataset used to train a single binary classifier. This approach produced a mean Matthews correlation coefficient (MCC) of 0.806 ± 0.017 SD across 100 cross-validation iterations. The 172 Level 3 pathways were predicted with an overall MCC of 0.726. Moreover, metabolite association with the 12 Level 2 pathway categories was predicted with an overall MCC of 0.891, representing significant transfer learning from the Level 3 pathway entries. These are the best metabolite pathway prediction results published so far in the field.

## 1. Introduction

Metabolism is the set of biochemical reactions occurring within organisms or individual cells that act to sustain life. These reactions are commonly represented as a metabolic network, i.e., a graph with metabolites as nodes and reactions as edges, with some edges having directionality. Parts of the metabolic network are defined as metabolic pathways based on many decades of study, research, and teaching of metabolism [1,2,3]. From these pathway definitions, metabolites are associated with specific pathways. Metabolic knowledgebases such as the Kyoto Encyclopedia of Genes and Genomes (KEGG) [4,5,6], MetaCyc [7], and Reactome [8] contain mappings between metabolites and associated pathways, but these mappings are incomplete and are very slow, tedious, and costly to discover. For this reason, accurately predicting the pathway involvement of detected metabolites would be a valuable capability across biological and biomedical research fields.

Various attempts have been made to develop machine learning models to predict metabolite pathway mappings given information about a metabolite’s chemical structure, especially pathway mappings for KEGG Level 2 (L2) ‘Metabolism’ pathway categories [9,10,11,12]. In several published attempts, the dataset used was discovered to be invalid [13]. Thus, Huckvale et al. created a new benchmark dataset of metabolite features and associated pathway mappings as labels, all derived from KEGG [14]. The features in this benchmark dataset were generated using an atom coloring technique developed by Jin and Moseley [15]. This work involved training a binary classifier for each pathway, the input being a representation of a metabolite and the output indicating whether or not the given metabolite is associated with the pathway [14]. Each model was only capable of predicting a single pathway. Through the use of a novel feature and pathway engineering approach, the later work of Huckvale and Moseley [16] created a single binary classifier capable of predicting an arbitrary metabolite-pathway mapping based on a pathway chemical structure representation. The novel feature and pathway engineering approach involves a cross-join of metabolite features with pathway features, creating a dataset that is 12 times larger than the original benchmark dataset: 68,196 metabolite-pathway pair entries vs. 5683 metabolite entries [16].

Collections of pathways can be organized into pathway categories and these categories can be organized into broader categories in a hierarchical manner. In the case of KEGG, pathways are organized into a hierarchy where the first level of the hierarchy includes ‘Metabolism,’ the top-level (Level 1 or L1) category containing metabolic pathways. The second level, i.e., L2, in the hierarchy includes 12 pathway categories, sometimes referred to as modules. These L2 pathway categories were the focus of past works [9,10,11,12,14,16]. However, there are 172 Level 3 (L3) pathways that are more granular. When combined with the broader 12 L2 pathway categories, a set of 184 metabolic “pathways” is created. In this situation, it is neither practical to train individual models for each pathway nor is it practical to train a multi-output model with 184 outputs because most of the resulting datasets would have too few positive entries for effective training. Having a significantly lower proportion of positive examples often results in excellent accuracy but poor recall, precision, and related metrics [17].

The cross-join technique introduced by Huckvale and Moseley [16] resolves these issues, since it is a single model with a single output and is capable of accepting an indefinite number of pathways into the training dataset. Specifically, the resulting dataset from the cross-join increases the number of positive examples to a level needed for effective training. Moreover, we show in this work that while the L3 pathways perform worse when trained by themselves, their performance increases when trained with the L2 pathway categories in the same dataset and with the same model. Likewise, the L2 pathways also increase in performance when trained alongside the L3 pathways. Combining both the broader pathway categories with the more granular individual pathways results in an improved overall model performance. This demonstrates that it is more than possible to train a model to reliably predict mappings to both L2 pathway categories and L3 individual pathways.

## 2. Materials and Methods

Our work began using the same methods as Huckvale et al. [14] to construct features representing metabolites using the atom coloring technique introduced by Jin and Moseley [15]. Each atom in a graph representing a chemical structure is “colored” by the bonded atoms around it, up to three bonds away. A feature vector of the count of atom colors present is created for each metabolite, and then these are summed to create similar feature vectors for pathways. Likewise, the molfiles and pathway mappings were obtained from KEGG using the kegg_pull Python package version 3.1.0 [18]. Then, the same cross-join technique, as introduced by Huckvale and Moseley [16], was used to construct the metabolite and pathway feature pairs. The metabolite-to-pathway mappings are defined by the KEGG pathway hierarchy available at https://www.genome.jp/kegg-bin/show_brite?br08901.keg (accessed on 1 June 2024). We see that the L1 pathway categories in the hierarchy include ‘Metabolism,’ and below that are the L2 pathway categories used in the prior works of Huckvale et al. [14,16] and other publications involving metabolic pathway prediction with KEGG [9,10,11,12]. At an additional hierarchy level below that, we see the L3 pathways are individual pathways as compared to categories of pathways in the hierarchy levels above. However, whether an individual pathway or a category of pathways, either can be defined as a collection of associated metabolites, enabling the construction of a cross-joined dataset of combined metabolite and pathway features, including both the L2 pathway categories and the L3 individual pathways. The L2 dataset was constructed by the prior work of Huckvale and Moseley [16]. In this work, we constructed the L3 dataset containing the metabolic pathways in the 3rd level of the KEGG hierarchy and the metabolites associated with them. Additionally, we constructed a 3rd combined dataset containing both L2 and L3 pathways. Table 1 provides details of these three datasets, including the total number of entries (metabolite pathway pairs) after the cross-join.

Both metabolite and pathway features were normalized using the softmax by bond count technique performed in Huckvale and Moseley [16]. This normalization technique groups each atom color feature of each entry by the bond count of the atom colors, i.e., 0, 1, 2, or 3. Then, the features are summed, providing a sum for each bond count, followed by the features being divided by their corresponding sum. While this is only an entry-wise normalization, in this work, we additionally normalized feature-wise using standard min/max scaling (each feature value ranges between 0 and 1). To compare how the scaling impacts model performance, we created both a scaled and non-scaled version of each of the L2 only, L3 only, and combined datasets, resulting in six total datasets. Note that the non-scaled L2-only dataset was part of the prior work of Huckvale and Moseley [16], and the five additional datasets were introduced in this work.

In the work of Huckvale and Moseley, the Multi-Layer Perceptron (MLP) model performed best and is also scalable, being able to perform just as well when training on the data in batches as opposed to current implementations of XGBoost [19], which are most optimal when trained on the entire dataset all at once [16]. This is a limitation when the dataset does not fit entirely into memory, necessitating batched training. Considering the superior performance and ability to scale to indefinitely large datasets, we only trained and evaluated MLP models in this work. Models were trained to perform binary classification, i.e., given both metabolite features and pathway features, predict whether or not the given metabolite is associated with the given pathway. Hyperparameters were tuned for each of the six subsets, producing six corresponding models. Note that the model trained on the non-scaled L2-only dataset was part of the prior work of Huckvale and Moseley [16], and the five additional models were introduced in this work. Models trained on the L2 datasets had an input layer size equal to the number of features, i.e., 14,655 + 5435 = 20,090. Likewise, the L3 models had an input size of 23,606, and the models trained on the combined dataset had an input size of 23,632. Being a binary classification problem, there was a single output layer that was converted to either 0 or 1 depending on whether it was above or below a binary classification threshold, which we tuned as a hyperparameter. The selected hyperparameters of the L3 and combined models, including hidden layer size and number of layers, are listed in Appendix A.

Models were evaluated using 10-fold cross-validation (CV) and stratified train test splits [20] over 100 CV iterations, with 10% of the entries sampled in the test sets. Only one test fold is utilized from each iteration. This evaluation methodology includes aspects of bootstrapping without replacement (i.e., a jackknife analysis with a 10% hold-out), which is superior for estimating distributions and statistics of model performance. However, reducing the number of CV iterations from 1000 to 100 was a pragmatic decision given the jump from training with 68,196 entries to 1,045,672 entries. Metrics used to evaluate the models included accuracy, precision, recall, F1 score, and Matthews correlation coefficient (MCC) [21]. While these metrics were used to compare the test sets’ predictions and ground truth, we additionally counted the individual true positives, true negatives, false positives, and false negatives of each individual pathway. This means that we calculated the metrics across all entries in the test set of each CV iteration, and we also split the test sets up by pathway category since each entry includes a pathway identity in the pair. We did not calculate these metrics for each pathway individually in each CV iteration because some of the pathways were too small, i.e., did not have a sufficient number of metabolites associated with them (positive examples) in order to consistently calculate valid scores (zero true/false positives can cause a division by zero when calculating MCC). Counting the individual number of true positives, false negatives, true negatives, and false negatives enabled us to sum these metrics across all 100 CV iterations and construct a single confusion matrix used to calculate the overall MCC per pathway. Since we were able to calculate metrics for the entire test set on each CV iteration, we were able to calculate means and standard deviations for the scores across all pathways. However, individual pathway scores do not have standard deviations since only a single MCC score can be calculated from the confusion matrix summed across all the CV iterations.

The hardware used for this work included machines with up to 64 gigabytes (GB) of random-access memory (RAM) and central processing units (CPUs) ranging from 3.5 gigahertz (GHz) to 3.6 GHz, each with 6 hyperthreaded (HT) cores. The CPU chips included ‘Intel(R) Core(TM) i7-6850K CPU@3.60GHz’ and ‘Intel(R) Core(TM) i7-5930K CPU@3.50GHz’. The graphic processing units (GPUs) used had 16 GB of GPU RAM, with the name of the GPU card being ‘NVIDIA GeForce RTX 4080′. The GPUs were sourced from the Nvidia corporation in Santa Clara, California, USA.

All code for this work was written in major version 3 of the Python programming language [22]. Data processing and storage were done using the Pandas version 1.0.3 [23], NumPy version 1.26.4 [24], and H5Py version 3.9.0 [25] packages. Models were constructed and trained using the PyTorch Lightning package version 2.1.1 [26] built upon the PyTorch package version 2.0.1 [27]. Hyperparameters were tuned using the Optuna package version 3.3.0 [28]. The optimal hyperparameters are listed in Appendix A. Metrics and the stratified train test splits were computed using the Sci-Kit Learn package version 1.3.0 [29]. Results were stored in an SQL database [30] using the DuckDB package version 1.0.0 [31]. Data visualizations were produced using the Tableau business intelligence software version 2024.2.2 [32] as well as the Seaborn package version 0.12.2 [33] built on the MatPlotLib package version 3.7.2 [34]. Statistical correlations were calculated using the SciPy package version 1.11.4 [35]. Computational resource usage of the training and evaluation was collected using the gpu_tracker package version 3.0.0 [36]. All code and data for reproducing these analyses can be accessed via the following Figshare item: https://doi.org/10.6084/m9.figshare.26505907 (accessed on 1 June 2024).

## 3. Results

### 3.1. Overall Model Performance across All Pathways

Figure 1 shows violin plots of the MCC scores across all 100 CV iterations for the L3 only and L2 + L3 combined datasets, separated by scaled and non-scaled features. Each MCC score was computed from all predictions in the test set of each CV iteration. As we can see, the L2 + L3 combined dataset has a higher median MCC and less variance than the L3-only dataset, whether the dataset was scaled or not. Scaling the features increased the median performance in both the L3 only and L2 + L3 combined datasets, respectively.

Table 2 shows the mean and standard deviation MCC from the results in Figure 1 and additionally includes those for the L2-only dataset. Appendix A contains scores for the other four metrics, i.e., accuracy, precision, recall, and F1 score. We see from Table 2 that while scaling the features improved the MCC of the L3-only and combined pathways; it decreased the MCC of the L2-only pathways. However, these MCCs are calculated from the entries available in the given dataset, with L2 entries missing from the L3-only dataset. So, the comparisons of these MCCs and violin plots in Figure 1 are not exactly apples-to-apples and oranges-to-oranges comparisons.

To address this shortcoming, Table 3 shows the MCC across only the L2 or L3 pathways within the L2 + L3 combined dataset. While Table 2 shows the MCC of the L2 pathways after training only on the L2 dataset and the L3 pathways after training on only the L3 dataset, Table 3 shows the overall MCC specific to L2 and L3 pathways, separating them after training on the L2 + L3 combined dataset. The MCCs in Table 3 were calculated by summing the true/false positives/negatives across all pathways of the given hierarchy level and across all CV iterations as described in the Methods, which prevents the calculation of a standard deviation. Appendix A contains these sums as well as the number of pathway/CV iteration pairs that were summed. Since there were 100 CV iterations and 12 L2 pathways, there were a total of 1200 counts that contributed to the L2 sums. The L3 pathway sums likewise had 17,200 counts. The summed counts were combined to construct confusion matrixes to calculate a single MCC for each combination of hierarchy levels and determine whether or not the features were scaled. We see that for both hierarchy levels, the pathway categories predicted significantly better when trained on the combined dataset than when trained separately. At the same time, the non-scaled features performed better for the L2 pathways when training on the L2 pathways alone (Table 2); scaling resulted in greater performance for the L2 pathways when trained on the L2 + L3 combined dataset (Table 3). Taking the best results between scaled vs. unscaled datasets, the L2-only trained model had a mean MCC of 0.784 vs. an overall MCC of 0.891 for the L2 + L3 trained model, representing a huge improvement! The improvement is well outside the standard deviations in Table 2, specifically the 0.013 standard deviation for the L2-only trained model and 0.021 standard deviation for the combined model, with the latter standard deviation treated as an upper limit for the L2 pathway overall MCC.

Table 4 shows the computational resource usage of the models trained on the L3 only and L2 + L3 combined datasets, separated by whether the features were scaled. These are the computational resources used when evaluating the models over the 100 CV iterations. We see that scaling the features resulted in slightly longer compute time but slightly less GPU RAM and RAM. For the scaled datasets, the L3-only dataset took longer to train and evaluate despite there being less RAM utilization. This is likely due to the models requiring more epochs to converge. From Huckvale and Moseley [16], L2-only training and evaluation using an unscaled MLP model took roughly 90 min for 1000 CV iterations. In comparison, the training and evaluation on the L2 + L3 combined unscaled dataset across 100 CV iterations took 131.1 h. A large time increase in training and evaluation is to be expected since the L2-only dataset has 68,196 entries versus 1,045,672 entries in the L2 + L3 combined dataset. Resource usage statistics were collected using the gpu_tracker package [36].

### 3.2. Model Performance Per Pathway in the Combined Dataset

The following results are based solely on models trained on the scaled L2 + L3 combined dataset. Appendix A contains the pathway information used to produce the following plots.

#### 3.2.1. Distribution of Pathway Statistics

There are two ways we measured the size of a pathway. The first is simply the number of metabolites associated with it. Each one of those metabolites has a certain number of non-hydrogen atoms as part of their molecular structure. So, the second way we measured a pathway size is the sum of all the non-hydrogen atoms across all the metabolites associated with the pathway. Figure 2 displays the distribution of the pathway sizes using both metrics.

Figure 3 shows the distribution of overall pathway-specific MCC scores across 100 CV iterations. While six pathways scored an overall MCC of less than 0.4 (see highlighted pathways in Appendix A), the majority of pathways scored above 0.4, with a median of 0.722.

#### 3.2.2. Comparing Pathway Category Size to MCC

Figure 4 shows scatterplots comparing the pathway size to its overall MCC score across 100 CV iterations. The X axes are shown both on the regular scale as well as on the log10 scale. Both forms of measuring pathway size are shown. The log10 scale shows a clear, nearly linear relationship between pathway size and MCC.

Table 5 shows the correlations between the size of the pathway and the overall MCC of the pathway, as plotted in Figure 4 above. While the correlation coefficients are moderate, the *p*-values are very low. We see that the correlation is higher and more statistically significant when measuring the pathway size as the sum of the number of non-hydrogen atoms across all compounds in the pathway as compared to the mere number of compounds in the pathway category. The correlation is even stronger when calculating the Pearson correlation coefficient on the log10 scale compared to Spearman on the regular scale.

Since we can calculate the overall MCC of an individual pathway across the CV iterations, we can also calculate the overall MCC across subsets of pathways by summing their confusion matrixes. Figure 5 shows the increase in MCC across the remaining pathways after filtering pathways below a size threshold from the MCC calculation. For example, a threshold size of 0 is the overall MCC across all pathways, and a threshold of 200 only calculates the MCC of pathways with at least 200 associated metabolites (Figure 5a). The same trend is shown when measuring pathway size by a total number of non-hydrogen atoms with thresholds ranging up to 5000 (Figure 5c). Filtering pathways from the calculation also results in fewer pathways to consider, and Figure 5b,d shows the drop in remaining pathways as the threshold increases. We see an increase in MCC up to above 0.88 when only considering the 25 largest pathways. These trends are expected and simply highlight the higher information content of larger pathways with respect to model training.

### 3.3. Impact of MCC When Filtering Pathways from the Training Set by Pathway Size Thresholds

To determine how including smaller pathways (pathways with less associated metabolites) in the training set impacts the performance of the larger pathways, we created additional subsets by filtering pathways below a size threshold from the combined and scaled subset. With a threshold of 3, no metabolites were filtered since the smallest pathways had at least three metabolites associated with them. Using a threshold of 5 resulted in a subset containing 181 out of the 184 combined pathways and so on, with thresholds of 7, 10, 12, 20, 50, and 100. This is distinct from the filtering in Figure 5 since those results were from filtering pathways from the MCC calculation, but the model was trained on all pathways. For the results in Figure 6, the model was trained on a subset of the pathways to determine how excluding pathways from the training set, not just the MCC calculation, impacts the performance of the 33 largest pathways. The largest threshold was 100, resulting in only 33 pathways remaining in its corresponding training set (Figure 6a). To make all of the filtering thresholds comparable, all MCCs reported are based on these 33 largest pathways. We see that pathways smaller than 12 hinder the performance of these 33 largest pathways, with the lowest performance of 0.853 with a filter of 5 versus the highest performance of 0.872 with the 12-metabolite threshold. However, filtering at 20 or above resulted in decreasing performance, starting at 0.851.

Since the 12-metabolite count threshold dataset performed the best, Figure 7 shows the distribution of MCC scores across CV iterations for the entirety of the test set containing 173 out of 184 original pathways. We see a 0.006 improvement (increase) in mean and a 0.004 improvement (decrease) in standard deviation compared to the scaled L2 + L3 combined dataset that contained all 184 pathways (Table 2).

## 4. Discussion

In this work, we demonstrate that it is possible to develop a single binary classifier that can predict not just KEGG L2 metabolic pathway categories but more granular L3 pathways, with MCCs mostly above 0.4 and reaching above 0.8 for the largest pathways. Like the work in Huckvale and Moseley [16], this single binary classifier approach uses a cross-join of metabolite and pathway features to create a much larger dataset for training and testing while also allowing prediction on arbitrary metabolites and pathways. This approach enabled the prediction of not just the 12 L2 pathways but the 184 L2 + L3 combined pathways by eliminating the complexity and limitations of training and deploying a binary classifier for every pathway category or a multi-output model with an unrealistically high number of outputs. In particular, a multi-output model would need to be retrained with each new pathway added.

Models trained only on L3 pathways did not perform as well (mean MCC of 0.655 ± 0.031 SD) as models trained only on L2 pathways (mean MCC of 0.784 ± 0.013 SD). Models trained on the L2 + L3 combined dataset outperformed all previous models with a mean MCC of 0.800 ± 0.021 SD. Moreover, when just looking at the performance of L2 pathways, the L2 + L3 combined model had an overall MCC of 0.891! These results clearly demonstrate that adding more pathways increases the overall performance of the model, especially for large pathways. These results are a clear example of transfer learning, where training on additional pathways improves the performance of other pathways. However, we observed a clear trend where smaller pathways perform worse than larger pathways. However, there appears to be a sweet spot around pathways with 12 metabolites and larger that add more information than noise during model training. A training set trimmed to pathways with 12 or more metabolites maximized the overall MCC of 0.872 for the 33 largest pathways as compared to trainsets with more or fewer pathways. This final 12-metabolite threshold dataset produced a mean MCC of 0.806 ± 0.017 SD. Therefore, we recommend that future datasets filter pathways that are too small from the training set in order to improve the performance of the remaining pathways. However, the model is still capable of predicting these smaller pathways if desired. We also observe an increase in performance when normalizing the data feature-wise in addition to entry-wise. Thus, we recommend min-max scaling for future datasets in addition to filtering by pathway size.

## 5. Conclusions

Despite their increased difficulty to predict, this work demonstrates that metabolite association with individual L3 pathways can be predicted with an overall MCC of 0.726. Moreover, metabolite association with L2 pathways can be predicted with an overall MCC of 0.891 using the L2 + L3 combined dataset, representing transfer learning with increased dataset size. In general, larger pathways perform better than smaller pathways, and the smallest pathways can even decrease the performance of larger pathways. The 12-metabolite threshold dataset produced the highest mean MCC of 0.806 ± 0.017 SD. Thus, filtering these smallest pathways can maximize model performance, but filtering too much can decrease the performance due to removing valuable information for training.

## Figures and Tables

**Figure 1 metabolites-14-00510-f001:**
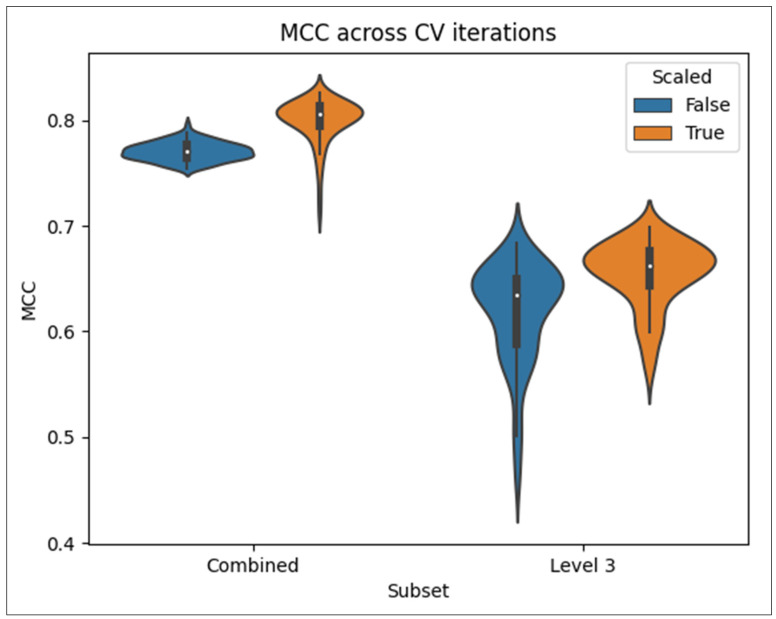
Violin plots of MCC scores across 100 CV iterations for models trained on the L3 only or L2 + L3 combined datasets, with or without scaling.

**Figure 2 metabolites-14-00510-f002:**
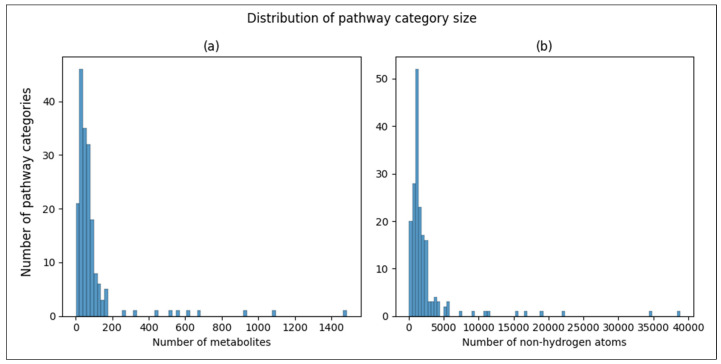
Histograms of pathway size. (**a**) Histogram of a number of metabolites associated with a pathway. (**b**) Histogram of number of non-hydrogen atoms summed across metabolites associated with a pathway.

**Figure 3 metabolites-14-00510-f003:**
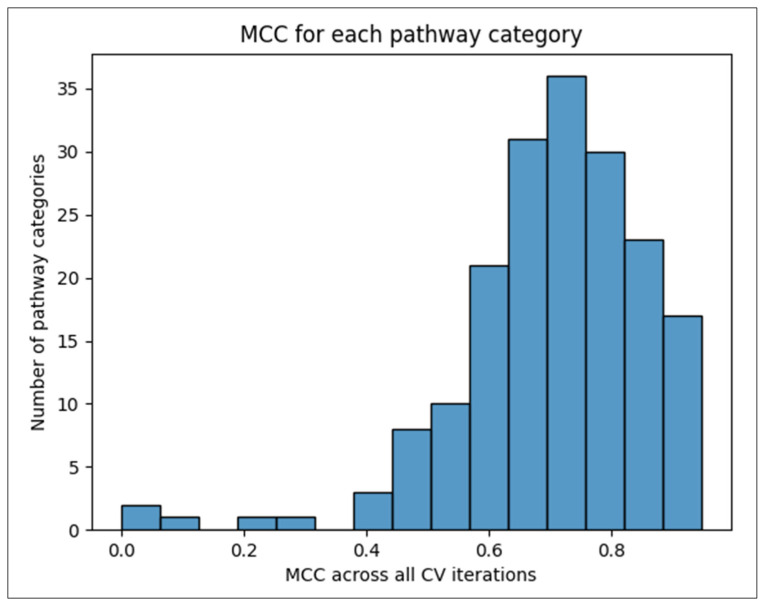
Histogram of pathway-specific overall MCC scores.

**Figure 4 metabolites-14-00510-f004:**
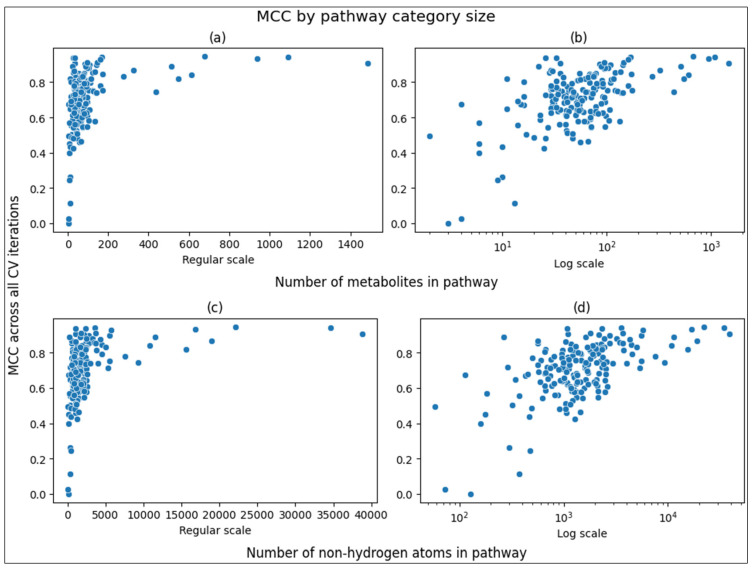
Scatterplots of overall MCC scores versus pathway size. (**a**,**b**) Scatterplot of overall MCC score versus number of metabolites associated with a pathway. (**c**,**d**) Scatterplot of overall MCC score versus number of non-hydrogen atoms summed across metabolites associated with a pathway.

**Figure 5 metabolites-14-00510-f005:**
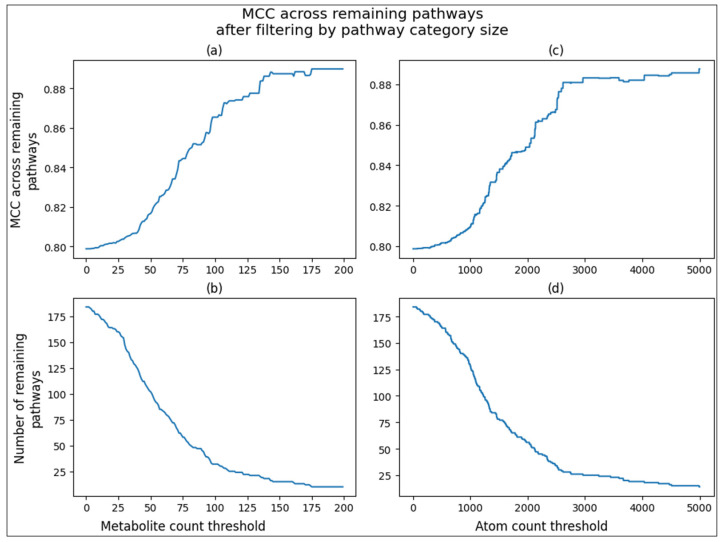
Line plots of overall MCC and pathway size based on pathway size filtering. (**a**) Line plot of overall MCC based on metabolite count filtering. (**b**) Line plot of number of pathways based on metabolite count filtering. (**c**) Line plot of overall MCC based on non-hydrogen atom count filtering. (**d**) Line plot of number of pathways based on non-hydrogen atom count filtering.

**Figure 6 metabolites-14-00510-f006:**
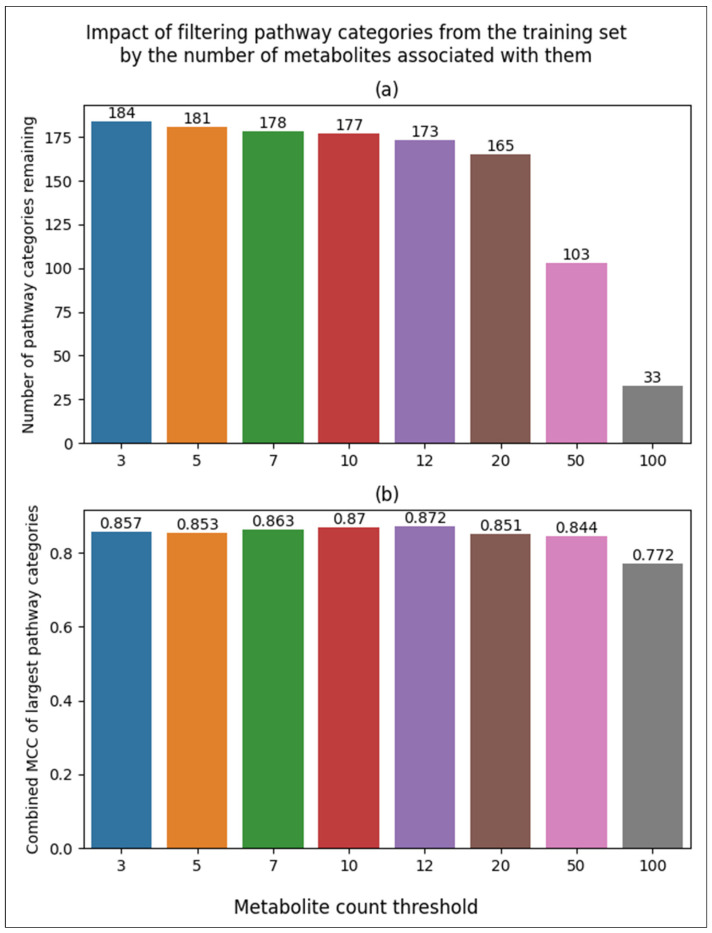
Bar plots of model performance based on pathway size filtering of the training set. (**a**) Bar plot of the number of pathways included in training based on metabolite count filtering. (**b**) Bar plot of overall MCC of the 33 largest pathways based on metabolite count filtering of the training set.

**Figure 7 metabolites-14-00510-f007:**
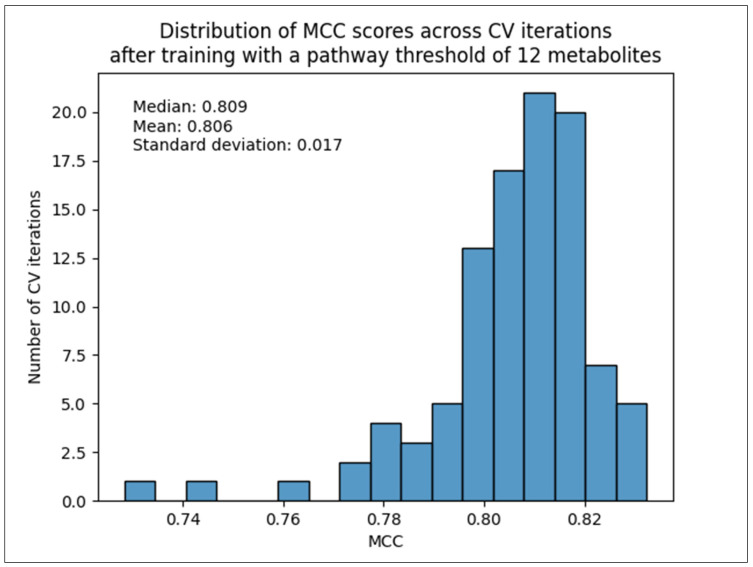
Histogram of MCC score across 100 CV iterations trained on the 12-metabolite count threshold dataset.

**Table 1 metabolites-14-00510-t001:** Description of cross-joined metabolite-pathway paired datasets used for model training and testing.

Dataset	# Metabolite Features	# Pathway Features	# Metabolites	# Pathways	# Entries
L2	14,655	5435	5683	12	68,196
L3	14,655	8951	5683	172	977,476
Combined	14,655	8977	5683	184	**1,045,672**

**Table 2 metabolites-14-00510-t002:** Mean model MCC performance metrics based on the given dataset.

Dataset	Scaled	Mean MCC	Standard Deviation
Combined	True	**0.800**	0.021
	False	0.771	0.009
L2 only	True	0.728	0.029
	False	0.784	0.013
L3 only	True	0.655	0.031
	False	0.618	0.048

**Table 3 metabolites-14-00510-t003:** Overall L2 + L3 combined model MCC performance metrics specific to pathway hierarchy level.

Test Set	Scaled	MCC
L2 only	True	**0.891**
	False	0.850
L3 only	True	0.726
	False	0.703

**Table 4 metabolites-14-00510-t004:** Hardware utilization for model training and evaluation across 100 CV iterations.

Dataset	Scaled	Resource	Unit	Amount
Combined	True	Compute time	Hours	131.8
		GPU RAM	Gigabytes	4.3
		RAM	Gigabytes	13.6
	False	Compute time	Hours	131.1
		GPU RAM	Gigabytes	4.6
		RAM	Gigabytes	14.1
L3 only	True	Compute time	Hours	171.3
		GPU RAM	Gigabytes	2.9
		RAM	Gigabytes	12.6
	False	Compute time	Hours	127.8
		GPU RAM	Gigabytes	4.1
		RAM	Gigabytes	16.6

**Table 5 metabolites-14-00510-t005:** Correlations between overall MCC and pathway size.

Pathway Size Metric	Correlation Coefficient/Scale	*p*-Value
# Compounds	0.442 Spearman/regular	3.463 × 10^−11^
# Non-hydrogen Atoms	0.501 Spearman/regular	4.234 × 10^−14^
# Compounds	0.572 Pearson/log10	2.177 × 10^−17^
# Non-hydrogen Atoms	0.584 Pearson/log10	3.160 × 10^−18^

## Data Availability

All code and data for reproducing these analyses can be accessed via the following Figshare item: https://doi.org/10.6084/m9.figshare.26505907.

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
