# Peer review of "Predicting the Association of Metabolites with Both Pathway Categories and Individual Pathways"

_metabolites, 2024, doi:10.3390/metabo14090510_

Round 1
Reviewer 1 Report
Comments and Suggestions for Authors
The manuscript presents a valuable contribution to the field of metabolomics and bioinformatics. The authors have effectively addressed the challenge of predicting metabolite-pathway associations using a machine learning approach. The study demonstrates the feasibility of using structural features and pathway information to accurately predict metabolite involvement in both broad pathway categories and specific pathways.
- Many studies like current study rely on publicly available datasets like those from KEGG, which might not capture the full spectrum of metabolic diversity found in different organisms, tissues, or disease states.
- Expand the dataset to include more diverse sources, such as metabolomics data from non-model organisms, various human tissues, and multiple disease conditions. This would help in training models that are more robust and generalizable across different biological contexts
- Sole reliance on metabolomics data may limit the understanding of metabolic pathway dynamics, which are influenced by transcriptional, translational, and post-translational modifications.
- Integrate multi-omics data, including transcriptomics, proteomics, and even genomics, to provide a holistic view of the metabolic pathways. This integration can enhance the predictive power of the models by incorporating regulatory and interacting networks that affect metabolite levels
o Basic feature sets derived directly from metabolomic data might not be sufficiently informative for pathway prediction
- Develop how to create better features for our analysis. Instead of just looking at whether a metabolite is there and how much, we should also think about how its amount changes in different situations, how it interacts with other things, and how it might be involved in metabolic pathways. One could also use information about the structure of these pathways and the relationships between different metabolites.
- Validation of predictive models often relies on computational cross-validation or independent datasets, which may not fully confirm the biological relevance of the predictions
- To test if the predicted metabolites are actually involved in the pathways, one could do experiments like targeted metabolomics or flux analysis. This means measuring specific metabolites or tracking how they flow through the pathways. We could also use CRISPR-Cas9 to turn genes on or off to see how that affects the levels of these metabolites.
Main strengths of the study are:
Big Data: The researchers combined different kinds of data to create a huge list of connections between metabolites and pathways. This helped their machine learning model learn a lot. Strong Methods: They used a reliable machine learning method and checked their results carefully to make sure they were accurate.
This study is well constructed, but there are a few things we could improve:
- This study used one database (KEGG) for information about metabolites and pathways. Using data from other sources, like databases for different kinds of organisms or diseases, could make our model work better for more things.
- Using data from other areas, like gene expression or protein levels, to get a better understanding of metabolic pathways and make better accurate predictions.
- This study could try more advanced machine learning methods, like deep learning or combining different models, to make this model perform even better and compare model evaluation.
- This study could find new ways to get useful information from the metabolite and pathway data, which could help us make more accurate predictions.
- This study should do experiments to see if our predictions are correct in real-life situations. This would give us stronger evidence that our findings are important.
Author Response
Reviewer 1:
The manuscript presents a valuable contribution to the field of metabolomics and bioinformatics. The authors have effectively addressed the challenge of predicting metabolite-pathway associations using a machine learning approach. The study demonstrates the feasibility of using structural features and pathway information to accurately predict metabolite involvement in both broad pathway categories and specific pathways.
Response:
We thank the reviewer for their positive comments!
Issue 1:
Many studies like current study rely on publicly available datasets like those from KEGG, which might not capture the full spectrum of metabolic diversity found in different organisms, tissues, or disease states.
- Expand the dataset to include more diverse sources, such as metabolomics data from non-model organisms, various human tissues, and multiple disease conditions. This would help in training models that are more robust and generalizable across different biological contexts
Sole reliance on metabolomics data may limit the understanding of metabolic pathway dynamics, which are influenced by transcriptional, translational, and post-translational modifications.
Response:
KEGG is a knowledgebase. It contains structured (machine readable) information and knowledge about genes, genomes, and metabolism along with other biologically related information. Such structured data is required to build a dataset for machine learning.
By virtue of utilizing all relevant KEGG compound and metabolic pathway entries, we are indirectly utilizing metabolic information for over 7000 organisms represented in KEGG Genome. However, the metabolic networks in KEGG are heavily E. coli centric, since it is the best studied organism with respect to metabolism.
Moreover, this manuscript presents the largest training set published so far, with over 1 million entries. The previous largest training set published had over 68,000 entries:
Erik D. Huckvale and Hunter N.B. Moseley. "Predicting The Pathway Involvement Of Metabolites Based on Combined Metabolite and Pathway Features" Metabolites 14, 266 (2024). https://doi.org/10.3390/metabo14050266
Before that, all prior published datasets and models were trained on less than 5600 entries. So from less than a year ago, we have increased the training set size by over a 100-fold, 178-fold to be more precise.
However, we do plan to expand the dataset size in the future. But that is beyond the scope of this manuscript.
Issue 2:
- Integrate multi-omics data, including transcriptomics, proteomics, and even genomics, to provide a holistic view of the metabolic pathways. This integration can enhance the predictive power of the models by incorporating regulatory and interacting networks that affect metabolite levels
Response:
We are building a tool to improve metabolomics data analysis and multi-omics integration. Specifically, we are building a machine learning model to predict metabolite association with a given metabolic pathway. Such a model would greatly increase the number of detected metabolites that can be mapped to human-defined metabolic pathways, which we hope to greatly improve the sensitivity and statistical power of pathway enrichment analyses.
In the future, we do plan to apply these machine learning models to demonstrate improvement to pathway enrichment analysis. But first, we much build the best predicting model possible and verify its predictive performance, which is main goal of this manuscript.
Issue 3:
o Basic feature sets derived directly from metabolomic data might not be sufficiently informative for pathway prediction
Response:
Respectably, we have already demonstrated that predicted annotations can be effectively used for annotation enrichment analysis in the follow paper:
Joshua M. Mitchell, Robert M. Flight, and Hunter N.B. Moseley. "Untargeted lipidomics of non-small cell lung carcinoma demonstrates differentially abundant lipid classes in cancer vs non-cancer tissue" Metabolites 11, 740 (2021). https://doi.org/10.3390/metabo11110740
To be clear, pathway enrichment analysis is a specific type of annotation enrichment analysis. So our goal is to demonstrate prediction performance that we already know is usable for annotation enrichment analysis.
Issue 4:
- Develop how to create better features for our analysis. Instead of just looking at whether a metabolite is there and how much, we should also think about how its amount changes in different situations, how it interacts with other things, and how it might be involved in metabolic pathways. One could also use information about the structure of these pathways and the relationships between different metabolites.
Response:
We have been developing tools for this purpose for a while. One example is Information-Content-Informed Kendall tau correlation:
Robert M Flight, Praneeth S Bhatt, and Hunter N.B. Moseley. "Information-Content-Informed Kendall-tau Correlation: Utilizing Missing Values" bioRxiv 2022.02.24.481854 (2022). https://doi.org/10.1101/2022.02.24.481854
But again, this is beyond the scope of the current manuscript.
Issue 5:
Validation of predictive models often relies on computational cross-validation or independent datasets, which may not fully confirm the biological relevance of the predictions
- To test if the predicted metabolites are actually involved in the pathways, one could do experiments like targeted metabolomics or flux analysis. This means measuring specific metabolites or tracking how they flow through the pathways. We could also use CRISPR-Cas9 to turn genes on or off to see how that affects the levels of these metabolites.
Response:
We agree that (simple) cross-validation may not fully confirm relevance of the predictions. However, we are doing a sophisticated evaluation methodology that utilizes a combination of bootstrap without replacement, jackknife analysis, and 10-fold cross validation. We are utilizing 100 independent 10-fold cross-validation iterations, where we only use one test fold from each iteration. We have added the following description to the methods to make this clearer:
“Models were evaluated using 10-fold cross validation (CV) and stratified train test splits (20) over 100 CV iterations with 10% of the entries sampled in the test sets. Only one test fold is utilized from each iteration. This evaluation methodology includes aspects of bootstrapping without replacement (i.e., a jackknife analysis with 10% hold out), which is superior for estimating distributions and statistics of model performance. However, reducing the number of CV iterations from 1,000 to 100 was a pragmatic decision given the jump from training with 68,196 entries to 1,045,672 entries.”
However, in the future, we do plan to utilize a large set of independent datasets to verify the utility of these predictions to improve pathway enrichment analysis. However, that is clearly beyond the scope of this manuscript.
Issue 6:
Main strengths of the study are:
Big Data: The researchers combined different kinds of data to create a huge list of connections between metabolites and pathways. This helped their machine learning model learn a lot. Strong Methods: They used a reliable machine learning method and checked their results carefully to make sure they were accurate.
Response:
We thank the reviewer for their positive comments!
Issue 7:
This study is well constructed, but there are a few things we could improve:
This study used one database (KEGG) for information about metabolites and pathways. Using data from other sources, like databases for different kinds of organisms or diseases, could make our model work better for more things.
Using data from other areas, like gene expression or protein levels, to get a better understanding of metabolic pathways and make better accurate predictions.
Response:
We have already addressed this issue above. In this manuscript, we demonstrate a 14-fold increase in dataset size from the prior largest dataset and over 178-fold increase in dataset size from any older publications.
Integration of other types of data is beyond the scope of the current manuscript.
Issue 8:
This study could try more advanced machine learning methods, like deep learning or combining different models, to make this model perform even better and compare model evaluation.
This study could find new ways to get useful information from the metabolite and pathway data, which could help us make more accurate predictions.
This study should do experiments to see if our predictions are correct in real-life situations. This would give us stronger evidence that our findings are important.
Response:
Again, these suggestions are clearly beyond the scope of a manuscript that demonstrates superior dataset size and performance from all prior publications, none of which utilized data from outside KEGG.
However, testing of other advanced machine learning methods is a logical next step we plan to do in the future.
Reviewer 2 Report
Comments and Suggestions for Authors
In this work, the authors developed MLP models to predict whether a metabolite is involved in a given Level 3 KEGG pathway. However, without sufficient context and explanations (which may have been provided in previous publications), this work appears more like an extension of prior methods, with the results feeling somewhat data-driven without deeper insights. Specifically, I have the following questions:
1. What are the metabolite features and pathway features mentioned in Table 1? A detailed description is required.
2. A more comprehensive explanation of the "softmax by bond count technique" mentioned in Lines 108-109 is needed.
3. In Lines 212-213, the conclusion cannot be made without providing the standard deviations.
4. In Lines 214-215, why does the model require more epochs to converge when using the L3 only scaled dataset?
5. In Line 238, which are the 5 pathways with MCCs less than 0.4? What is their size, and why do they have low MCCs?
Comments on the Quality of English LanguageThe quality of English is acceptable.
Author Response
Reviewer 2:
In this work, the authors developed MLP models to predict whether a metabolite is involved in a given Level 3 KEGG pathway. However, without sufficient context and explanations (which may have been provided in previous publications), this work appears more like an extension of prior methods, with the results feeling somewhat data-driven without deeper insights. Specifically, I have the following questions:
Response:
We demonstrate predictions on both L2 and L3 KEGG metabolic pathways. And, of course, we are building on prior published methods, especially the following paper:
Erik D. Huckvale and Hunter N.B. Moseley. "Predicting The Pathway Involvement Of Metabolites Based on Combined Metabolite and Pathway Features" Metabolites 14, 266 (2024). https://doi.org/10.3390/metabo14050266
But this is the FIRST demonstration of predicting metabolite association with L3 KEGG pathways. All prior publications only predicted L2 pathway associations. Most of those prior publications only predicted on 11 of the 12 KEGG L2 pathways; however, all of our prior publications predicted across all 12 L2 KEGG pathways.
Moreover, we increased the dataset size to over one million entries, a 14-fold increase in dataset size from the prior largest dataset and over 178-fold increase in dataset size from any older publications. Performance on the L2 pathways is the best demonstrated so far in the field.
Issue 1:
1. What are the metabolite features and pathway features mentioned in Table 1? A detailed description is required.
Response:
We thank the reviewer for pointing this out! It is sometimes hard to step back and see a manuscript from a new reader’s perspective. We have added the following details about the metabolite and pathway features:
“Our work began using the same methods as Huckvale et al [14] to construct features representing metabolites using the atom coloring technique introduced by Jin and Moseley [15]. Each atom in a graph representation of a chemical structure is “colored” by the bonded atoms around it, up to three bonds away. A feature vector of the count of atom colors present is created for each metabolite and then these are summed to create similar feature vectors for pathways.”
Issue 2:
2. A more comprehensive explanation of the "softmax by bond count technique" mentioned in Lines 108-109 is needed.
Response:
Again, thanks for pointing out this deficiency in our methods description! We have made the following improvements:
“Both metabolite and pathway features were normalized using the softmax by bond count technique performed in Huckvale and Moseley [16]. This normalization technique groups each atom color feature of each entry by the bond count of the atom colors i.e. 0, 1, 2, or 3. Then the features are summed, providing a sum for each bond count, followed by the features being divided by their corresponding sum”
Issue 3:
3. In Lines 212-213, the conclusion cannot be made without providing the standard deviations.
Response:
We are assuming the reviewer is referring to the following statement:
“Taking the best results between scaled vs unscaled datasets, the L2 only trained model had a mean MCC 0.784 vs an overall MCC of 0.891 for the L2+L3 trained model, representing a huge improvement!”
Due to the way we had to calculate the overall MCC, we cannot calculate a standard deviation. However, we can use the standard deviations for the mean MCC in Table 2 as an estimate. The L2-only trained model has a mean MCC of 0.784 ± 0.013 and the mean MCC for the combined dataset is 0.800 ± 0.021. We can treat the 0.021 standard deviation for the combined dataset as an upper limit for the standard deviation for the L2 pathway overall MCC of 0.891. Clearly, the improvement is real. We have added this rationale to make it clearer to the reader:
“Taking the best results between scaled vs unscaled datasets, the L2 only trained model had a mean MCC 0.784 vs an overall MCC of 0.891 for the L2+L3 trained model, representing a huge improvement! The improvement is well outside the standard deviations in Table 2, specifically the 0.013 standard deviation for the L2 only trained model and 0.021 standard deviation for the combined model, with the latter standard deviation treated as an upper limit for the L2 pathway overall MCC.”
Issue 4:
4. In Lines 214-215, why does the model require more epochs to converge when using the L3 only scaled dataset?
Response:
We do not know, but we suspect the epoch loss persistently improved just enough to prevent triggering early stopping, since we used a rather overly stringent early stopping criteria. As far as why particularly the L3 scaled dataset required more epochs to converge using the same criteria, it could be that the scaling allowed for greater improvement, but the high variance in predictability of the L3 pathways resulted in more epochs in order to see that improvement.
Issue 5:
5. In Line 238, which are the 5 pathways with MCCs less than 0.4? What is their size, and why do they have low MCCs?
Response:
These pathways are very small as indicated in Figure 4. But we thank the reviewer for bringing this issue up, since we had miscounted. There are actually 6 pathways with MCCs less than 0.4. The sixth one had an MCC of 0.3986, which is likely why we had missed it. We have highlighted these pathways in Table S4 in supplemental material and added the following in the main text:
“While 6 pathways scored an overall MCC of less than 0.4 (see highlighted pathways in Supplemental Table S4), the majority of pathways scored above 0.4, with a median of 0.722.”
Reviewer 3 Report
Comments and Suggestions for Authors
The authors developed a machine learning model that successfully predicts metabolite associations with granular KEGG Level 3 metabolic pathways. The results demonstrate that model achieved high accuracy and demonstrated significant improvements in metabolite-pathway prediction. These results represent the best published outcomes in this area of research.
Therefore, the manuscript is recommended for publication.
Author Response
Reviewer 3:
The authors developed a machine learning model that successfully predicts metabolite associations with granular KEGG Level 3 metabolic pathways. The results demonstrate that model achieved high accuracy and demonstrated significant improvements in metabolite-pathway prediction. These results represent the best published outcomes in this area of research.
Therefore, the manuscript is recommended for publication.
Response:
We thank the reviewer for their very positive comments!
We have worked very hard to greatly increase the dataset size to over 1 million entries, which has greatly improved machine learning performance.
Reviewer 4 Report
Comments and Suggestions for Authors
In this manuscript, a study has been carried out for the estimation of road involvement of metabolites in both road categories and individual roads. The references considered in the study are from recent research in the literature. However, the following points should be clarified for the study to better contribute to researchers in this field.
1. More detailed information about the multi-layer perceptron model used in the study should be provided.
2. Detailed information on how the parameters and hyper-parameters of the model under consideration are set.
3. Since the dataset considered in the study is quite large, sufficient information about why a deep learning model was not used in the study should be provided.
Author Response
Reviewer 4:
In this manuscript, a study has been carried out for the estimation of road involvement of metabolites in both road categories and individual roads. The references considered in the study are from recent research in the literature.
Response:
We thank the reviewer for their positive comments. We have tried to reference recent relevant research.
Issue 1:
However, the following points should be clarified for the study to better contribute to researchers in this field.
1. More detailed information about the multi-layer perceptron model used in the study should be provided.
Response:
We have added the following details about the MLP models:
“Models trained on the L2 datasets had an input layer size equal to the number of features i.e. 14,655 + 5,435 = 20,090. Likewise, the L3 models had an input size of 23,606 and the models trained on the combined dataset had an input size of 23,632. Being a binary classification problem, there was a single output layer which was converted to either 0 or 1 depending on whether it was above or below a binary classification threshold which we tuned as a hyperparameter. The selected hyperparameters of the L3 and combined models, including hidden layer size and number of layers, is listed in Supplemental Table S1.”
Issue 2:
2. Detailed information on how the parameters and hyper-parameters of the model under consideration are set.
Response:
We used the Optuna package for hyper-parameter tuning:
“Hyperparameters were tuned using the Optuna package [28].”
We have added a table of optimal hyperparameters used to generate the models to supplemental material:
“The optimal hyperparameters are listed in Supplemental Table S1.”
Issue 3:
3. Since the dataset considered in the study is quite large, sufficient information about why a deep learning model was not used in the study should be provided.
Response:
The term “deep learning” is poorly defined; but, we understand the point the reviewer is trying to make.
First, we wanted comparability to prior datasets and models to demonstrate the improvement from the dataset via transfer learning.
Second, it is unclear what deep learning methods would provide even equivalent let alone superior results. Our atom coloring method is generating atom-centric neighborhood (subgraph) representations that can be distinctly combined for long distance information. It is unclear if graph spectral or spatial network representations can effectively capture what our atom coloring features can. Graph attention network may be able to, but it will take quite a bit of time to determine this.
Third, we felt the improvements in the dataset and resulting models would provide timely new information and resources to the field. This is the first demonstration of predicting pathway association at finer grained metabolic pathways.
Needless to say, we do plan on exploring other graph based deep learning methods in the future.
Round 2
Reviewer 2 Report
Comments and Suggestions for Authors
The authors have addressed my questions. I have no further questions.
Comments on the Quality of English Language
The quality of English language is acceptable.